# Safety and Efficacy of Regorafenib and 5-Fluorouracil Combination Therapy in Refractory Metastatic Colorectal Cancer After Third-Line Treatment: An Institutional Experience

**DOI:** 10.3390/biomedicines13051151

**Published:** 2025-05-09

**Authors:** Maen Abdelrahim, Abdullah Esmail, Ebtesam Al-Najjar, Bayan Khasawneh, Godsfavour Umoru, Waseem Abdelrahim, Karen Abboud, Veronica B. Ajewole

**Affiliations:** 1Section of GI Oncology, Houston Methodist Neal Cancer Center, Houston Methodist Hospital, Houston, TX 77094, USA; aesmail@houstonmethodist.org (A.E.);; 2Michael E. DeBakey HS for Health Professions, Houston, TX 77030, USA

**Keywords:** regorafenib plus 5-FU, 5-fluorouracil, refractory mCRC, multi-kinase inhibitor, OS

## Abstract

**Background**: Colorectal carcinoma (CRC) is one of the most common cancer types along with breast, prostate, and lung cancer. Many patients with CRC present with metastatic disease despite receiving standard first- and second-line therapies; thus emerges the demand for implementing new therapies that could improve outcomes among CRC patients. This case series was conducted to assess the efficacy and safety of regorafenib plus 5-fluorouracil (5-FU) in patients with refractory metastatic CRC (mCRC). **Methods**: We conducted a retrospective analysis of data from adult patients aged 18 and above who were diagnosed with refractory mCRC and received regorafenib plus 5-FU combination therapy at Houston Methodist Hospital between November 2017 and October 2023. Our study focuses on assessing key outcomes, including Overall Survival [OS], Progression-Free Survival [PFS], and safety. **Results**: Among the 12 patients we included in this study who underwent regorafenib plus 5-FU combination therapy for refractory mCRC after receiving at least three prior lines of treatment, the best response for six patients (50%) was successfully achieved, with disease control within 7–12 weeks from therapy initiation. Patients had an overall good tolerance for this treatment regimen and reported only the most common adverse events, including Hand-Foot Syndrome (HFS), mucositis, and hypertension (HTN), which were mostly resolved with dose adjustment of medications. **Conclusions**: This study highlights that using a combination of regorafenib plus 5-FU can be a potential treatment option for patients with refractory mCRC. Additional research, including prospective clinical trials, is required to assess the effectiveness and safety of regorafenib and 5-FU combination therapy in comparison to other currently limited treatment options.

## 1. Introduction

Colorectal carcinoma (CRC) is one of the most diagnosed cancers around the world. It affects both men and women, but the numbers vary slightly between genders [1]. For men, CRC is the third most prevalent cancer after lung and prostate cancers. This means that out of all the men diagnosed with cancer, about 10% of them have CRC. That is approximately 746,000 cases. Among women, CRC is the second most prevalent cancer after breast cancer. Thus, about 9.2% of all women diagnosed with cancer have CRC. That is approximately 614,000 cases [1].

In the United States, CRC is the third most diagnosed and the third leading cause of cancer-related deaths for both men and women. However, it ranks second in cancer-related deaths overall and is the leading cause of death among men younger than 50 years. More than half of all cases and deaths can be linked to modifiable risk factors including smoking, unhealthy diet, high alcohol consumption, physical inactivity, and excess body weight. CRC stands as the second most prevalent cause of cancer-related deaths in the United States. Every three years, the American Cancer Society (ACS) releases updated statistics on CRC, extracting data on incidence from population-based cancer registries and mortality data from the National Center for Health Statistics (NCHS). In 2023, it is projected that approximately 153,020 individuals will be diagnosed with CRC, while 52,550 will die from the disease. Among those cases, 19,550 will affect individuals under the age of 50, resulting in 3750 deaths [2]. Among people diagnosed with mCRC, approximately 70% to 75% of patients survive beyond 1 year, 30% to 35% beyond 3 years, and fewer than 20% beyond 5 years from diagnosis. The treatment strategy for mCRC is termed resectable when the primary tumor and all metastases are amenable to complete surgical removal. However, in these patients, nodal infiltration and occult micro-metastatic dissemination are common. Resection of mCRC achieves long-term cure for less than 20% of mCRC patients [3]. Currently, several different drugs are being used in mCRC treatment, with first- and second-line options including the systemic drugs 5-FU, irinotecan, oxaliplatin, bevacizumab, cetuximab, panitumumab, and the oral drug capecitabine, as well as different combinations of these drugs, such as the FOLFOX regimen (5-FU, leucovorin, and oxaliplatin), the FOLFIRI regimen (5-FU, leucovorin, and irinotecan), and the XELOX regimen (capecitabine and oxaliplatin), either with or without a monoclonal antibody agent [4]. However, treatment options beyond the third line of refractory mCRC remain challenging, and despite the availability of multiple options, outcomes are generally poor [5].

Regorafenib is an orally active, potent multi-kinase inhibitor that targets a broad range of angiogenic, stromal, and oncogenic kinases, including Vascular Endothelial Growth Factor (VEGF) receptors 1, 2, and 3, tyrosine kinase with immunoglobulin and epidermal growth factor homology domain 2 (TIE-2), platelet-derived growth factor receptor-*β*, c-kit, ret, raf-1, and BRAF [6,7].

5-FU is an antimetabolite drug that has been widely used since 1957 to treat different types of cancer, including CRC and breast cancer [8]. 5-FU is a pyrimidine analogue that acts as an antimetabolite of uracil, causing cell death. After entering the cell, 5-FU is converted to one of several active metabolites. These metabolites, in turn, inhibit the enzymes thymidylate synthase and uracil-DNA-glycosylase, interfering with DNA synthesis and repair, respectively. In addition, one of the 5-FU metabolites is incorporated into RNA, thereby disrupting its processing and function [9].

In September 2012, the US Food and Drug Administration (FDA) approved regorafenib for the treatment of mCRC patients who had failed FOLFIRI chemotherapy regimens; an anti-VEGF pathway therapy; and an anti-Epidermal Growth Factor Receptor (EGFR) therapy (for KRAS wild-type patients). In the phase III CORRECT (Regorafenib Monotherapy for Previously Treated mCRC) trial data demonstrated improved OS benefit for mCRC patients treated with regorafenib versus placebo in patients with treatment-refractory mCRC (6.4 vs. 5.0 months; *p* = 0.0052) [10].

In recent years, numerous studies and hypotheses have been explored to combine regorafenib with various drugs, aiming to establish and validate its potential synergistic effects. These investigations have been driven by the pursuit of enhanced therapeutic outcomes, particularly in the context of treatment-refractory mCRC. One of these hypotheses is that combining regorafenib with anti-programmed cell death 1 (PD-1)/anti-programmed cell death ligand 1 (PD-L1), antibodies may be associated with significant clinical benefit in patients with mCRC. The median PFS and OS were 3.6 months [95% confidence interval (CI), 1.8–5.4] and 10.8 months (95% CI, 5.9–NA), respectively [11].

In other retrospectively analyzed patients with advanced or mCRC who received at least one dose of immune checkpoint inhibitors ICIs combined with regorafenib in 14 Chinese medical centers, the median PFS was 3.1 months (95% CI, 2.3–4.2) and the median OS was 17.3 months [12].

While prior studies explored regorafenib with fluoropyrimidines, this study uniquely provides real-world dosing optimization and outcomes in a rare, off-label refractory mCRC population, addressing a critical gap in practical guidance for heavily pretreated patients. This study aims to evaluate the efficacy and safety of regorafenib combined with intravenous 5-FU in heavily pretreated patients with mCRC who have progressed beyond third-line therapy. Additionally, it seeks to provide practical insight on dosing and toxicity management for this off-label combination, addressing a critical gap in real-world data.

## 2. Materials and Methods

### 2.1. Study Design and Participants

This case series evaluates the treatment outcomes for a total of 12 mCRC patients. The inclusion criteria for the study were pathology-confirmed adenocarcinoma of the colon or rectum with radiological evidence of mCRC, age of at least 18 years, patients who had received at least three lines of previous treatment for mCRC, and Eastern Cooperative Oncology Group Performance Status (ECOG PS) score restricted to 0–1. Data were collected for patients treated at Houston Methodist Cancer Center between November 2017 and October 2023. The study cohort of 12 patients (2017–2023) includes 7 patients previously reported in Haque et al. (2017–2021) with extended follow-up, plus 5 additional patients enrolled subsequently [5].

We collected demographic information, including age, gender, ECOG PS, primary site of CRC, metastatic sites (liver and lung), primary tumor surgical interventions and metastasectomies, prior adjuvant chemotherapy, number of previous chemotherapy sessions, initial dose of regorafenib, and presence of KRAS mutations. We gathered information on adverse events associated with regorafenib, including Hand-Foot Syndrome (HFS), Hypertension (HTN), skin rash, and instances of emergency hospitalization. Adverse event severity was assessed based on the National Cancer Institute Common Terminology Criteria for Adverse Events (NCI-CTCAE). Good tolerance was defined as completing at least two treatment cycles without grade 3/4 toxicities requiring permanent discontinuation, maintaining a regorafenib dose of at least 80 mg/day, and avoiding hospitalization due to treatment-related adverse events. These data were retrospectively gathered from Electronic Medical Records (EMR). Approval for this study was granted by the Institutional Review Board (IRB) of the Houston Methodist Research Institute.

### 2.2. Patients’ Characteristics

A total of 12 patients with mCRC were included in this study, including eight males and four females, with an average age of 59 years old. Ethnically, seven of the patients were of Caucasian race, four were of Asian race, and one was of black race. Patients’ age ranged from 31–60 years old at the time of diagnosis and from 40–65 years old at therapy initiation. Most of the enrolled patients had HTN as a co-morbid condition. Seven patients (58%) had left-sided CRC, two patients (16%) had right-sided CRC, and three patients (25%) had rectal cancer. The primary tumor was resected in eleven patients (91%). Nine patients (75%) had stage IV at the time of diagnosis, two patients (16%) had stage III, and one patient had stage I. All patients had mutated KRAS or NRAS except for one patient and none had microsatellite instability.

The prevalent metastatic site was confirmed through imaging such as Computed Tomography (CT) and Magnetic Resonance Imaging (MRI) scans. The liver was identified as the primary location (in all 12 patients) followed by the lung (in 8 patients). All patients received (FOLFOX) chemotherapy and anti-VEGF therapy prior to combination therapy; three patients started with FOLFOX, two with capecitabine and oxaliplatin, and one with capecitabine monotherapy. The second-line chemotherapy for most patients comprised (FOLFIRI) and bevacizumab. Additionally, two patients received regorafenib as monotherapy prior to the use of combination therapy. Other commonly used therapies include trifluridine/tipiracil, ramucirumab, and capecitabine. The information on therapies used prior to regorafenib plus 5-FU combination therapy is listed in Table 1.

### 2.3. Treatment and Assessment

The clinical outcomes of interest encompassed the optimal response to the combined therapeutic regimen involving regorafenib and 5-FU. Safety data were available for the twelve patients who received each regimen.

Regorafenib is typically prescribed at a standard dose of 160 mg (administered as four 40 mg tablets) once a day. This treatment follows a schedule of three weeks on, followed by one week off therapy. Because of the adverse events of regorafenib, the standard doses were not applicable to every patient. Most patients in our study were initiated on a regorafenib dosage of 80 mg/day, a dose that was subsequently escalated within one to two weeks to attain a target dosage of 120 mg. Further adjustments were made, allowing for a maximum dosage of 160 mg, determined by the individual’s tolerability.

Regarding 5-FU, dosages were mostly administered as a 400 mg/m^2^ bolus, followed by a continuous infusion of 2400 mg/m^2^ over 46 h, initiated on the first day of the therapy.

Key endpoints included PFS, OS, and the assessment of adverse events. PFS was defined as the time span between the initiation of combination therapy and the occurrence of clinically substantiated evidence indicating disease progression. On the contrary, OS was described as the timeframe from the initiation of treatment to mortality resulting from any cause.

We kept detailed records of the given doses of 5-FU and regorafenib when treatment started to the point of discontinuation or the time of the last follow-up, as shown in Figure 1. Through this approach, we aimed to evaluate the therapeutic efficacy, survival outcomes, and safety with combined treatment administration.

## 3. Results

### 3.1. Efficacy

Our study yielded promising outcomes among the twelve patients whose tumor responses were evaluated, showcasing disease control (i.e., Partial Response (PR) or Stable Disease (SD)) in six patients. A positive outcome was observed in 50% of patients. Notably, one patient (patient #2) exhibited a PR, while the remaining five patients demonstrated SD status, which were the best responses recorded to this therapy.

Optimal responses were observed within the timeframe of 7–12 weeks after the initiation of the therapeutic regimen. Unfortunately, six patients (50%) exhibited progression of the disease within 6–8 weeks of treatment initiation, requiring a transition to an alternative therapeutic approach. With follow-up duration ranging from 2 to 20 months, seven out of the twelve patients remained alive. A comprehensive summary of the clinical outcomes for all twelve patients is presented in Table 2.

In our study, we observed that the median OS is 12 months. The 10 months OS of 61% (95% Confidence Interval (CI):25–84%) and 12 months OS of 50% (95% CI: 18.8–75.3%) OS Kaplan–Meier analysis is illustrated in Figure 2. Additionally, the median PFS was 3.15 months.

### 3.2. Safety

Most of the patients in our study were initiated on a regorafenib dosage of 80 mg/day, a dose that was subsequently escalated within one to two weeks to attain a target dose of 120 mg, with further adjustment to a maximum of 160 mg as deemed tolerable. Notably, six patients (50%) initiated therapy with a dose of 80 mg, five (41%) received an initial dose of 120 mg, and one patient (8%) embarked on treatment with a dose of 160 mg. Three patients (25%) required further de-escalation to maintain tolerability at 80 mg.

Remarkably, only one patient exhibited tolerance to the 160 mg dosage but developed pneumonitis thereafter. In terms of 5-FU dosages, the majority, ten patients (83%) precisely, received a 400 mg/m^2^ bolus and a continuous infusion of 2400 mg/m^2^ over a 46 h duration, starting from the first day of therapy. However, four patients (33%) required a 20% dose reduction in 5-FU to enhance tolerability, and one patient received treatment in an external medical facility.

Overall, seven (58%) out of twelve patients had a good tolerance to treatment, defined as completing at least two cycles without grade 3/4 toxicities requiring discontinuation or hospitalization. These included Patients #2, #4, and #8–#12, who maintained regorafenib at 120 mg without severe adverse events, and Patient #1, who required dose reduction to 80 mg but continued therapy.

The least favorable response was in patient #5, who initially started regorafenib at a dose of 80 mg, with subsequent escalation to 120 mg. However, the patient developed grade 3 HFS. This adverse event necessitated the transition to an alternative therapeutic approach and a modified treatment plan.

In the case of patient #7, the initiation of regorafenib as monotherapy preceded the subsequent addition of 5-FU. Unfortunately, this patient experienced regorafenib-related pneumonitis, which entailed home oxygen support and discontinuation of the combined regimen. The patient experienced improvement with post-treatment care which included a course of steroids with a tapering regimen.

During treatment, three patients (25%) encountered mild adverse events that required adjustments to regorafenib dosages. For instance, patients #1 and #6 exhibited grade 2 HFS and grade 1–2 mucositis, respectively, prompting a modification of the regorafenib dosage from the initial target dose of 120 mg to 80 mg daily. Patient #3, with a pre-existing history of HTN, experienced hypertensive urgency marked by a systolic blood pressure exceeding 180 mmHg. In response, adjustments were made to the patient’s antihypertensive medications, resulting in the resolution of hypertensive urgency without recurrence. The remaining patients exhibited tolerance to the medication at the intended goal dose of 120 mg.

Additionally, gastrointestinal toxicities were closely monitored due to their prevalence with regorafenib and 5-FU. Mucositis (grade 1–2) occurred in patient #6, starting on day 7 of cycle 1 and resolving within 1 week after reducing regorafenib to 80 mg/day and using supportive oral treatment. Notably, no cases of diarrhea were reported in this cohort, potentially due to proactive use of antidiarrheal agents (e.g., loperamide) initiated at treatment onset or the lower starting dose of regorafenib (80–120 mg).

Adverse events, including HFS, hypertension, and pneumonitis, were managed with tailored interventions: topical steroids and dose reductions for HFS, antihypertensives for hypertension, and steroids with supplemental oxygen for pneumonitis, as detailed in Table 1 and Table 3. These strategies, optimized for this off-label regimen, ensured a 75% continuation rate. A comprehensive summary of treatment doses and associated adverse events for all patients is available for review in Table 1 and Table 3.

## 4. Discussion

Certainly, regorafenib offers promising outcomes for patients with refractory mCRC who have had disease progression despite all approved standard therapies. Clinical data indicates that regorafenib tends to yield better results in patients with good PS; therefore, it should be incorporated into the management of mCRC before patients become too frail and begin to experience a rapid decrease in PS. In our study, we highlight that using a combination of regorafenib plus 5-FU could be an optimal option for patients with refractory mCRC; through our investigation of treatment efficacy and safety, we provide insights into the potential benefits of this combination therapy, offering a valuable option for patients facing this challenging condition.

A study conducted by Fabien Calcagno et al. at Franche-Comté Cancer Hospitals included 29 patients who were treated with regorafenib monotherapy for mCRC, with doses ranging from 80 to 120 mg. In the CRC retrospective study, the median OS was six months and the median PFS was not assessed [13].

In our study, one patient (8.3%) achieved a PR, five patients (41.6%) experienced SD, and six patients (50%) had progression of disease. The median PFS was 96 days, (3.15) months. The 10-month OS was 61% (95% Confidence Interval (CI):25–84%) and the 12-month OS was 50% (95% CI: 18.8–75.3%). These findings indicate improved outcomes compared to previous research findings, suggesting that combining regorafenib and 5-FU leads to better results than using regorafenib as a monotherapy.

In our study, we observed impressive outcomes in two patients with mCRC who had previously undergone multiple lines of chemotherapy. The first patient was a 48-year-old male who was treated with an initial regorafenib dose of 80 mg and escalated to 120 mg. The patient was able to receive 12 cycles of regorafenib plus 5-FU therapy, with five prior chemotherapies consisting of capecitabine + oxaliplatin, FOLFIRI + bevacizumab, FOLFOX, trifluridine/tipiracil + bevacizumab, and pembrolizumab monotherapy. The patient began regorafenib as a sixth-line chemotherapy after undergoing sigmoid colectomy. MRI scan results showed no signs of tumor recurrence, indicating PR with stable liver metastasis until he had progression of disease. The other patient was a 59-year-old man with multiple lung and liver metastases who received seven prior chemotherapies and was initiated on regorafenib and 5-FU as the eighth line of therapy. Following the administration of two courses of regorafenib, the patient showed a decrease in CEA levels from 382 to 291 ng/mL, reached 177 ng/mL after two courses of therapy, and eventually experienced disease progression. This observation suggests a potential therapeutic benefit of regorafenib plus 5-FU therapy in controlling disease progression and reducing tumor burden in heavily pretreated patients with refractory mCRC.

Similarly, in another study conducted by Marks et al., two patients with mCRC who were resistant to first-line FOLFOX and second-line FOLFIRI were administered regorafenib in conjunction with either capecitabine or 5-FU. In both cases, despite prior treatment failures, the patients demonstrated evidence of disease control. One patient-maintained SD for at least one month with regorafenib and capecitabine therapy, while the other achieved at least two months of SD before starting to accumulate new metastatic foci. Additionally, in vitro studies revealed synergistic effects when regorafenib was combined with 5-FU, suggesting a potential mechanistic basis for the observed clinical responses. Comparing the outcomes of our study with those reported by Marks et al., we observe similarities in patient characteristics and treatment responses. Both studies highlight the potential efficacy of regorafenib plus 5-FU in refractory mCRC. The observed disease control and prolonged survival in some patients underscore the importance of exploring novel treatment combinations in this challenging patient population [14].

In our study, which investigates the combination therapy of regorafenib plus 5-FU in mCRC patients, we found that most adverse events associated with the regimen were of low severity, indicating a manageable safety profile within our patient cohort. However, a notable proportion of three patients (25%) required adjustments to the regorafenib dosage based on clinical considerations, with two patients discontinuing treatment due to intolerable adverse effects.

Our findings also revealed variations in the initial doses chosen in clinical practice, with 80 mg (n = 50%) being the most selected dose, followed by 120 mg (n = 41.6%) and 160 mg (n = 8.3%). Notably, a direct relationship was observed between the initial dose and the frequency of Grade 3 or 4 adverse events; as the initial dose increases, a higher frequency of Grade 3/4 adverse events was observed (0% for 80 mg, 8.3% for 120 mg, and 8.3% for 160 mg). Interestingly, these observations regarding dose-dependent toxicity align with recent findings from the ReDOS study, which demonstrated that a strategy of weekly dose escalation of regorafenib from 80 mg to 160 mg/day was non-inferior to a starting dose of 160 mg/day for survival outcomes. This suggests that a gradual dose-escalation approach may offer comparable efficacy while potentially reducing the incidence of severe adverse events [15]. The absence of diarrhea in this cohort contrasts with studies like Zanwar et al. [16], where 65% of patients experienced grade 3/4 toxicities, including diarrhea. This may reflect our use of lower initial regorafenib doses (80–120 mg) and proactive supportive care, such as loperamide. The single case of mucositis was effectively managed with dose reduction and oral rinses, highlighting the importance of early intervention to maintain treatment continuity.

However, our study’s findings diverge significantly from those reported by Zanwar et al. [16] in a study conducted at a tertiary cancer center in India. Their investigations involved 23 patients treated with regorafenib; dose reduction was required for 86.9% of patients. Thirteen patients were initiated at a lower dose of 120 mg initially due to the poor tolerance of the 160 mg dose observed in the first ten patients. The occurrence of Grade 3/4 drug-related adverse events was notably higher, with at least one Grade 3/4 toxicity noted in 65% of cases studied [16].

This comparison highlights differences in treatment outcomes and adverse event profiles between our study and the study by Zanwar et al. [16], underscoring the importance of considering regional and patient-specific factors when determining treatment approaches for mCRC patients receiving regorafenib therapy. Overall, our study underlines the importance of personalized dosing and vigilant monitoring of adverse events in clinical practice, aiming to maximize treatment efficacy while minimizing treatment-related toxicity.

Further collaborative research efforts are needed to elucidate the optimal dosing strategies and improve outcomes for patients with CRC undergoing regorafenib plus 5-FU therapy. Hence, it is imperative to closely monitor patients during the initial course of regorafenib treatment to proficiently address and mitigate any potential treatment-related adverse effects.

Furthermore, our study delved into the examination of patients’ status during their initial follow-up visit after the commencement of regorafenib plus 5-FU therapy. Conducting vigilant surveillance at these junctures facilitates the early identification of any adverse effects associated with the treatment, thus allowing for timely intervention before any exacerbation in severity. Prior clinical investigations have also underscored the favorable tolerability and moderate antitumor efficacy of an initial regorafenib dose ranging from 80 mg to 120 mg, particularly when employed as salvage therapy for mCRC [17].

The promising 50% DCR and 12-month OS in this rare cohort directly supported funding for a Phase II trial at Houston Methodist Neal Cancer Center (NCT06887218, HMCC-GI24-001), approved in February 2025, with a planned enrollment of 56 patients. Despite its small sample size and retrospective design, this study’s success as a foundational step validates its role in driving prospective research. The constrained sample size reflects the off-label use of the treatment in refractory mCRC, limiting eligibility to a rare and highly specific patient subgroup. To confirm these findings and optimize treatment sequencing and combination strategies for refractory mCRC, larger prospective studies are essential.

## 5. Conclusions

The findings of our study suggest that regorafenib and 5-FU combination therapy is a possible treatment option in mCRC patients, especially when considering systemic therapy beyond the third line. This regimen warrants proactive evaluation against other salvage therapies. Further prospective studies with larger cohorts are essential to confirm these findings and optimize the efficacy and safety of this treatment for refractory mCRC patients.

## Figures and Tables

**Figure 1 biomedicines-13-01151-f001:**
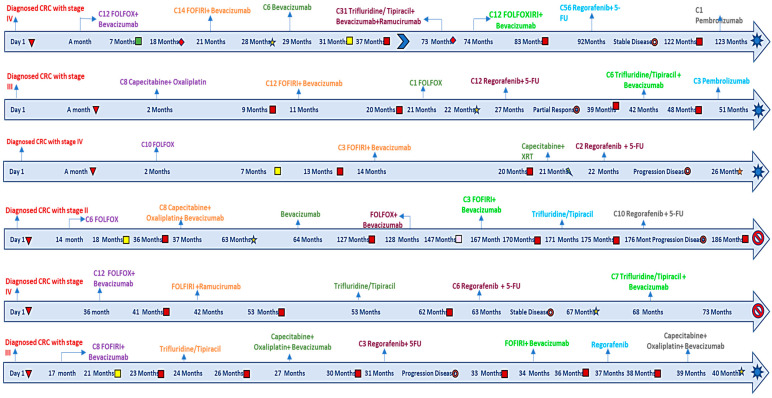
Detailed timelines for regorafenib and 5-FU patients.

**Figure 2 biomedicines-13-01151-f002:**
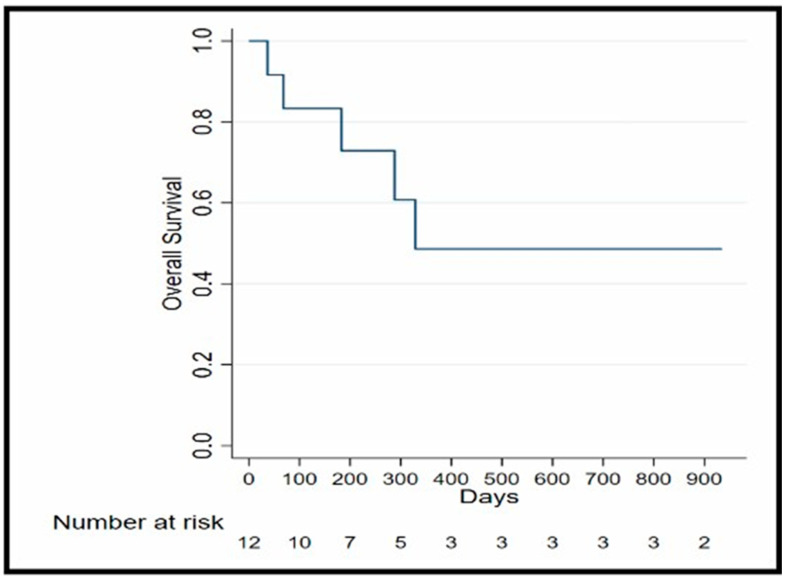
OS Kaplan–Meier analysis.

**Table 1 biomedicines-13-01151-t001:** Basic characteristics of patients.

Patient	Age at Therapy Initiation (Gender)	Diagnosis (Stage at Diagnosis)	Chemotherapies Prior to Rego + 5-FU (Including Maintenance Therapy)	History of Tumor Resection	Mutations	Sites of Metastasis	Comorbidities
**1**	65 (Female)	Left-sided colon cancer. (IV)	FOLFOX + bevacizumab FOLFIRI + bevacizumab bevacizumab monotherapy Trifluridine/Tipiracil initially with bevacizumab and then Ramucirumab FOLFIRINOX + bevacizumab	Yes	KRAS-mutant (G12D) NRAS-neg BRAF-neg MSI-stable	Peritoneum, ovaries, abdominal wall, liver	Osteoarthritis
**2**	48 (Male)	Left-sided colon cancer. (III)	Capecitabine + Oxaliplatin Capecitabine monotherapy FOLFIRI + bevacizumab FOLFOX + Oxaliplatin	Yes	KRAS-mutated MYC-mutated TP53-mutated NTRK 1–3-neg	Liver	Hypertension
**3**	58 (Male)	Right-sided colon cancer (IV)	FOLFOX FOLFIRI + bevacizumab Restarted FOLFIRI + bevacizumab.	Yes	KRAS-mutated TP53-mutated PIK3C-mutated BRAF-neg NTRK 1–3-neg MSI-stable HER-2-neg NRAS-neg	Liver and lung	Hypertension Diabetes Mellitus
**4**	46 (Male)	Left-sided colon cancer. (II)	FOLFOX Capecitabine + oxaliplatin + bevacizumab Capecitabine + bevacizumab bevacizumab FOLFOX + bevacizumab bevacizumab + 5-FU FOLFOX + bevacizumab FOLFIRI + bevacizumab Trifluridine/Tipiracil	Yes	NRAS mutation MSI-stable, FAP with pathogenic APC variant. HER2 negative.	Liver, lungs, bone, and peritoneum	None
**5**	63 (Female)	Left-sided colon cancer (IV)	FOLFOX + bevacizumab FOLFIRI + ramucirumab FOLFIRI + aflibercept Trifluridine/Tipiracil 5-FU	Yes	NRAS-mutated KRAS-neg BRAF-neg MSI-Stable	Liver and lung	Hypothyroidism Nephropathy
**6**	58 (Male)	Left-sided colon cancer (IV)	FOLFOX FOLFIRI + bevacizumab Trifluridine/Tipiracil Capecitabine + oxaliplatin + bevacizumab	Yes	-BRAF-neg -KRAS-mutated -NRAS-neg -HRAS-neg -MSI-Stable	Liver and lung	Hypertension Hypercholesterolemia
**7**	59 (Male)	Rectal cancer (IV)	Capecitabine FOLFOX capecitabine + bevacizumab irinotecan + bevacizumab bevacizumab panitumumab + irinotecan cetuximab + irinotecan 5-FU + irinotecan + cetuximab regorafenib monotherapy	Yes	KRAS wild type MSI stable	Liver and lung	Hypertension Gout
**8**	46 (Female)	Rectal cancer (IV)	FOLFOX + bevacizumab FOLFIRI + bevacizumab FOLFIRI + cetuximab	No	BRAF D594G pMMR PIK3CA MSI stable	Lung, Liver, Bone, Supraclavicular Lymph nodes	Hypertension
**9**	51 (Male)	Right-sided colon cancer (IV)	Capecitabine + oxaliplatin FOLFOX + bevacizumab FOLFIRI + bevacizumab capecitabine + bevacizumab bevacizumab + irinotecan + oxaliplatin	Yes	P53 MSS	Liver and Lung	Hypertension
**10**	52 (Male)	Rectosigmoid cancer (IV)	FOLFOX + bevacizumab FOLFIRI + cetuximab	Yes	FGFR1 MYC MSS	Liver, Lung and Peritoneum	Hypertension
**11**	60 (Male)	Rectal cancer (IV)	FOLFOX + bevacizumab FOLFIRI + cetuximab	No	HRAS G13C pMMR	Liver and Lymph nodes	Hypertension, Diabetes Mellitus, Asthma, Chronic Kidney Disease, Stroke
**12**	40 (Female)	Sigmoid cancer (IV)	FOLFOX + bevacizumab FOLFIRI + cetuximab	Yes	PTEN pMMR MSI stable	Liver and Peritoneum	Hypertension

**Table 2 biomedicines-13-01151-t002:** Outcomes in patients receiving regorafenib and 5-FU.

Patient #	Best Response (Time to the Best Response From Initiation in Weeks)	Progression or Discontinuation of Therapy (Time to Progression/Therapy Discontinuation)	Therapy after Rego + 5-FU	Time to the Last Follow-Up (in Months)	Status at Last Follow-Up
**1**	Stable disease (130)	Progression (130)	Pembrolizumab	31	Alive on other therapy
**2**	Partial response (53)	Progression (53)	Trifluridine/Tipiracil + bevacizumab pembrolizumab	30	Alive on other therapy
**3**	Progressive disease (7)	Progression (7)	Nivolumab + regorafenib Trifluridine/Tipiracil + bevacizumab Tolfenamic acid	20	Alive on other therapy
**4**	Progressive disease (8)	Progression (30)	Pembrolizumab + regorafenib	9.6	Deceased
**5**	Stable disease (17)	Discontinuation—Toxicity (17)	Trifluridine/Tipiracil + bevacizumab	11	Deceased
**6**	Progressive disease (13)	Progression (13)	FOLFIRI bevacizumab + oxaliplatin + capecitabine	11.5	Alive on other therapy
**7**	Stable disease (7)	Discontinuation—Toxicity (7)	None	2	Deceased
**8**	Progressive disease (13)	Progression (13)	Trifluridine/Tipiracil + bevacizumab	4.5	Alive on other therapy
**9**	Progressive disease (8)	Progression (8)	Trifluridine/Tipiracil + Bevacizumab	8	Alive on other therapy
**10**	Stable disease (6)	Progression (6.5)	Panitumumab + 5-FU	6	Deceased
**11**	Progressive disease (4)	Progression (4)	None	1	Deceased
**12**	Stable disease (7)	Discontinuation (7)	None	5	Alive

**Table 3 biomedicines-13-01151-t003:** Dosing and reported adverse events for patients receiving regorafenib and 5-FU.

Patient #	Regorafenib Dose at Initiation	Regorafenib Dose at Last Follow-Up or Discontinuation	5-FU Dose at Initiation (mg/m^2^)	5-FU Dose at Last Follow-Up or Discontinuation (mg/m^2^)	Adverse Events Reported
**1**	120 mg	80 mg	Day 1: 400 Day 2: 2400	Day 1: 320 Day 2: 1920	Grade 1–2 HFS
**2**	80 mg	120 mg	Received in an outside facility	Received in an outside facility.	Well-tolerated
**3**	120 mg	120 mg	Day 1: 400 Day 2: 2400	Day 1: 400 Day 2: 2400	Grade 3 HTN
**4**	120 mg	120 mg	Day 1: 400 Day 2: 2400	Day 1: 400 Day 2: 2400	Well-tolerated
**5**	120 mg	80 mg	Day 1: 400 Day 2: 2400	Day 1: 320 Day 2: 1920	Grade 3 HFS
**6**	120 mg	80 mg	Day 1: 400 Day 2: 2400	Day 1: 320 Day 2: 1920	Grade 1–2 mucositis managed with regorafenib dose reduction to 80 mg and oral rinses, resolved in 7 days.
**7**	160 mg	160 mg	Day 1: 320 Day 2: 1920	Day 1: 320 Day 2: 1920	Grade 3 pneumonitis
**8**	80 mg	120 mg	Day 1: 400 Day 2: 2400	Day 1: 400 Day 2: 2400	Well-tolerated
**9**	80 mg	120 mg	Day 1: 400 Day 2: 2400	Day 1: 400 Day 2: 2400	Well-tolerated
**10**	80 mg	120 mg	Day 1: 400 Day 2: 2400	Day 1: 400 Day 2: 2400	Well-tolerated
**11**	80 mg	120 mg	Day 1: 400 Day 2: 2400	Day 1: 400 Day 2: 2400	Well-tolerated
**12**	80 mg	120 mg	Day 1: 400 Day 2: 2400	Day 1: 400 Day 2: 2400	Well-tolerated

## Data Availability

Data are contained within the article.

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
