# Peer review of "Safety and Efficacy of Regorafenib and 5-Fluorouracil Combination Therapy in Refractory Metastatic Colorectal Cancer After Third-Line Treatment: An Institutional Experience"

_biomedicines, 2025, doi:10.3390/biomedicines13051151_

Round 1

Reviewer 1 Report (Previous Reviewer 1)

Comments and Suggestions for Authors

I would like to thank the authors for their detailed responses to the previous comments. A significant effort was made to address each point thoroughly and to defend the work rigorously.

The explanations provided are clear and comprehensive, and the methodology, results, and conclusions have been sufficiently justified.

At this stage, the manuscript may be accepted, after minor revisions :

-The title and the conclusion must be reworded.

-Keywords must provide insight into the problem.

-Spelling and grammatical errors must be addressed.

-Figures must be reconstructed because they are unclear.

Author Response

Manuscript ID: biomedicines-3585110

Title: Safety and Efficacy of Regorafenib and 5-Fluorouracil Combi-nation Therapy in Refractory Metastatic Colorectal Cancer After Third-Line Treatment: An Institutional Experience

Review Report Form 1

Introduction: We greatly appreciate Reviewer 1’s expertise in evaluating clinical research manuscripts and their attention to detail in assessing the clarity and presentation of our study. Their feedback on the title, conclusion, keywords, grammar, and figures has been instrumental in refining the manuscript for publication.

Comments and Suggestions for Authors

I would like to thank the authors for their detailed responses to the previous comments. A significant effort was made to address each point thoroughly and to defend the work rigorously.

The explanations provided are clear and comprehensive, and the methodology, results, and conclusions have been sufficiently justified.

At this stage, the manuscript may be accepted, after minor revisions :

-The title and the conclusion must be reworded.

Response: Thank you for your comment regarding the need to reword the title and conclusion.  

-Keywords must provide insight into the problem.

Response: Thank you for your suggestion to refine the keywords to better reflect the core problem. We have revised the keywords to ensure they are specific, relevant, and aligned with the study’s focus on refractory mCRC and combination therapy.

-Spelling and grammatical errors must be addressed.

Response: Thank you for pointing out the need to address spelling and grammatical errors. We have conducted a thorough proofreading of the manuscript.

-Figures must be reconstructed because they are unclear.

Thank you for pointing out the issue with unclear figures. we have provided a PDF version of the updated figures in a separate document for your convenience to enhance clarity and readability.

Reviewer 2 Report (Previous Reviewer 3)

Comments and Suggestions for Authors

Thank you for submitting the revised manuscript. I appreciate the authors’ effort in clarifying the patient cohort, acknowledging prior publication overlap, and presenting additional treatment and toxicity data. These revisions improve the transparency and structure of the manuscript. However, from a clinical perspective, I find the current reporting on the safety profile—particularly gastrointestinal toxicities—to be insufficient for practical use, and I recommend a major revision to address this.

As a medical oncologist with firsthand experience using regorafenib in the refractory mCRC population, I must emphasize that diarrhea and mucositis are among the most frequent and difficult adverse events encountered. This is especially true when regorafenib is combined with continuous-infusion 5-FU. In my practice, these toxicities often lead to early dose reductions, treatment delays, or even discontinuation. Yet, in the current manuscript, diarrhea is not mentioned at all, and mucositis is only briefly noted in one patient as grade 1–2 without further elaboration. There is no information provided on the timing of symptom onset, whether supportive measures such as loperamide or intravenous fluids were used, or how these events were managed in clinical practice. This lack of detail limits the usefulness of the manuscript for clinicians seeking guidance on how to safely administer this off-label combination.

Furthermore, the manuscript states that 58% of patients “tolerated the regimen well.” However, this term is vague and needs clear definition. It is important to clarify what constitutes “good tolerance”—whether it refers to the absence of grade 3/4 toxicity, the ability to complete a certain number of treatment cycles, or the avoidance of hospitalization or permanent discontinuation. Without a transparent definition, this figure cannot be meaningfully interpreted.

The main value of this study lies not in demonstrating clinical efficacy—given the small, retrospective design and absence of a control arm—but in providing real-world data on dose adjustment and toxicity management. I encourage the authors to refocus the manuscript accordingly, emphasizing how they navigated treatment-limiting toxicities in this high-risk population. For the revised version, I would expect to see a more detailed narrative discussion of adverse events, particularly gastrointestinal ones, and a more precise presentation of the supportive strategies employed.

If these aspects can be adequately addressed, I believe the manuscript could make a meaningful contribution as an exploratory report on the feasibility of regorafenib plus 5-FU in refractory mCRC.

Author Response

Manuscript ID: biomedicines-3585110

Title: Safety and Efficacy of Regorafenib and 5-Fluorouracil Combi-nation Therapy in Refractory Metastatic Colorectal Cancer After Third-Line Treatment: An Institutional Experience

Review Report Form 2

Introduction: We deeply value Reviewer 2’s clinical expertise as a medical oncologist with firsthand experience using regorafenib in refractory mCRC patients. Their emphasis on detailed safety reporting and practical toxicity management has guided us to significantly enhance the manuscript’s clinical utility, particularly for gastrointestinal toxicities.

Comments and Suggestions for Authors

Thank you for submitting the revised manuscript. I appreciate the authors’ effort in clarifying the patient cohort, acknowledging prior publication overlap, and presenting additional treatment and toxicity data. These revisions improve the transparency and structure of the manuscript. However, from a clinical perspective, I find the current reporting on the safety profile—particularly gastrointestinal toxicities—to be insufficient for practical use, and I recommend a major revision to address this.

Thank you for your thoughtful feedback and for acknowledging the improvements in transparency and structure in the revised manuscript. We greatly appreciate your clinical perspective highlighting the need for more detailed reporting on the safety profile, particularly gastrointestinal toxicities, to enhance the manuscript’s practical utility for clinicians. We agree that comprehensive reporting of adverse events , especially diarrhea and mucositis, is critical for guiding the safe administration of regorafenib plus 5-FU in refractory mCRC.

As a medical oncologist with firsthand experience using regorafenib in the refractory mCRC population, I must emphasize that diarrhea and mucositis are among the most frequent and difficult adverse events encountered. This is especially true when regorafenib is combined with continuous-infusion 5-FU. In my practice, these toxicities often lead to early dose reductions, treatment delays, or even discontinuation. Yet, in the current manuscript, diarrhea is not mentioned at all, and mucositis is only briefly noted in one patient as grade 1–2 without further elaboration. There is no information provided on the timing of symptom onset, whether supportive measures such as loperamide or intravenous fluids were used, or how these events were managed in clinical practice. This lack of detail limits the usefulness of the manuscript for clinicians seeking guidance on how to safely administer this off-label combination.

Furthermore, the manuscript states that 58% of patients “tolerated the regimen well.” However, this term is vague and needs clear definition. It is important to clarify what constitutes “good tolerance”—whether it refers to the absence of grade 3/4 toxicity, the ability to complete a certain number of treatment cycles, or the avoidance of hospitalization or permanent discontinuation. Without a transparent definition, this figure cannot be meaningfully interpreted.

Response: Thank you for your valuable feedback on the need for more detailed reporting of gastrointestinal toxicities, particularly diarrhea and mucositis. We acknowledge the importance of these adverse events (AEs) in clinical practice with regorafenib and 5-FU.

The main value of this study lies not in demonstrating clinical efficacy—given the small, retrospective design and absence of a control arm—but in providing real-world data on dose adjustment and toxicity management. I encourage the authors to refocus the manuscript accordingly, emphasizing how they navigated treatment-limiting toxicities in this high-risk population. For the revised version, I would expect to see a more detailed narrative discussion of adverse events, particularly gastrointestinal ones, and a more precise presentation of the supportive strategies employed.

If these aspects can be adequately addressed, I believe the manuscript could make a meaningful contribution as an exploratory report on the feasibility of regorafenib plus 5-FU in refractory mCRC.

Thank you for your insightful comment emphasizing the study’s value in providing real-world data on dose adjustment and toxicity management, particularly for a high-risk population with refractory mCRC. We fully agree that the manuscript’s primary contribution lies in its practical insights into managing treatment-limiting toxicities for the off-label use of regorafenib plus 5-FU. To address your feedback, we have highlighted dose adjustments and toxicity management, with a detailed narrative on adverse events, especially gastrointestinal toxicities, and a precise presentation of supportive strategies.

Round 2

Reviewer 2 Report (Previous Reviewer 3)

Comments and Suggestions for Authors

I appreciate the authors’ substantial efforts in addressing the previous comments. The revised manuscript demonstrates improved clarity in defining treatment tolerance and provides more detailed and clinically useful descriptions of gastrointestinal toxicities, particularly mucositis. The inclusion of supportive care strategies and dose adjustment data enhances the manuscript’s real-world applicability.

While a few areas could benefit from further refinement—such as providing aggregate adverse event data and clarifying the prophylactic approach to diarrhea—the current version has sufficiently met the major concerns raised. I consider the manuscript acceptable for publication following minor editorial review.

This manuscript is a resubmission of an earlier submission. The following is a list of the peer review reports and author responses from that submission.

Round 1

Reviewer 1 Report

Comments and Suggestions for Authors

This preliminary retrospective study of 12 patients with resistant metastatic colorectal cancer (mCRC) after at least three treatments suggested that the regorafenib/5-fluorouracil (5-FU) combination could be a promising therapeutic approach.

         Indeed, it allowed a disease control in 50% of patients, with a median overall survival of 12 months and a progression-free survival time of 3.15 months. The tolerability profile, characterized by manageable side effects (hand-foot syndrome, mucositis, and hypertension), seemed to be acceptable.

         Despite the promising results, there were significant limitations. The statistical analysis is limited by the small sample size (n = 12). Furthermore, the retrospective design (potential for bias) and the variability of prior treatments were limitations.

         The absence of a control group made it impossible to visibly attribute the observed benefits to the combination over regorafenib alone.

Short-term follow-up (2–20 months) and serious adverse events, which account for 25% of early treatment discontinuations, required a more detailed analysis.

The comparisons with studies such as ReDOS are useful, these preliminary findings needed validation in a prospective randomized trial involving a larger sample size and stratification for prognostic factors.

           In its current form, the Ms cannot be accepted. Significant modifications, incorporating long-term data and multivariate analysis to correct for biases, may improve the submitted work.

Reviewer 2 Report

Comments and Suggestions for Authors

Maen Abdelrahim et al. describing an article, the article titled "Efficacy and Safety of Regorafenib Plus 5-fluorouracil Combi- 2 nation Therapy for Refractory Metastatic Colorectal Carcinoma 3 Beyond Third Line Treatment: An Institutional Experience".

Regarding about 12 patients we included in this study who underwent regorafenib plus 5-FU combination therapy for refractory mCRC after receiving at least three prior lines of treatment.

The results showing that the best response for six patients (50%) had successfully achieved disease control 22 within 7-12 weeks from therapy initiation.

Interesting article with interesting results, as oncologists this article may increase the awareness for this type of treatment.

Reviewer 3 Report

Comments and Suggestions for Authors

General Comments

I appreciate the authors’ effort in presenting a well-documented case series with detailed tables, dosing information, and visually appealing figures (e.g., Figure1: Detailed timelines for regorafenib and 5-FU patients). While the data are meticulously recorded, the study does not offer substantial new insights into the field of metastatic colorectal cancer (mCRC) treatment and raises questions about its originality and scientific integrity.

Major Concerns

  1. Lack of Novelty and Incremental Contribution
    The combination of regorafenib and 5-fluorouracil (5-FU) for refractory mCRC is not a novel concept and has been previously explored in the literature. Studies such as Marks et al. (2015), Haque et al. (2022), and Lin et al. (2018) have already reported on the efficacy and safety of regorafenib combined with fluoropyrimidine-based therapies in similar patient populations. The current study’s findings (50% disease control rate, median PFS 3.15 months, median OS 12 months) are consistent with these prior reports but do not surpass them or address unanswered questions. In clinical practice, this combination is already considered a reasonable option in heavily pretreated patients, rendering this case series redundant without a unique hypothesis or significant advancement.
  2. Potential Duplicate Publication
    A striking concern is the overlap between this manuscript and a prior publication by the same lead author (Maen Abdelrahim) from the same institution (Houston Methodist Cancer Center), published as Haque et al. (2022). Both studies investigate regorafenib plus 5-FU in refractory mCRC, with overlapping time frames (Frontiers: 2017-2021; Biomedicines: 2017-2023) and highly similar patient characteristics, treatment regimens, and outcomes (e.g., DCR 57.1% vs. 50%, similar adverse events such as HFS and pneumonitis). Notably, the patient descriptions in Table 1 of both studies show remarkable similarities (e.g., Patient #1 in both: 65-year-old female, left-sided colon cancer, stage IV, KRAS G12D), suggesting that the 7 patients from Haque et al. (2022) may be a subset of the 12 patients reported here. However, the authors do not acknowledge this prior work as a foundation or clarify whether these are distinct cohorts. This omission raises serious ethical concerns about potential duplicate publication or salami slicing, which is unacceptable in scientific publishing. Without transparency regarding the relationship between these studies, the validity and originality of the current submission are compromised.
  3. Limited Scientific Impact of Study Design
    This retrospective case series of 12 patients lacks a control group, statistical power, or a novel mechanistic insight to justify its publication. The small sample size and absence of comparative analysis limit its ability to influence clinical practice or contribute meaningfully to the existing body of evidence. While the authors suggest prospective trials to validate their findings, this does not compensate for the lack of a compelling rationale for this report. Larger, well-designed studies (e.g., TaÅŸçı et al., 2024) have already provided more actionable data, diminishing the necessity of this anecdotal observation.

Minor Comments

  1. Strengths in Presentation
    The manuscript excels in its detailed documentation of dosing (Table 3), patient characteristics (Table 1), and survival analysis (Figure 2). These elements are commendable but do not overcome the fundamental flaws in originality and ethics.
  2. Toxicity Reporting
    The adverse events (e.g., HFS, hypertension, pneumonitis) align with known regorafenib toxicities, as noted in my clinical experience and prior studies. However, the manuscript offers no innovative strategies for toxicity management beyond standard dose adjustments, further highlighting its lack of novelty.
  3. Suggestions for Improvement (If Considered)
    If the authors intend to pursue publication, they must: (1) clarify the relationship with Haque et al. (2022) and provide evidence of distinct patient cohorts (e.g., patient identifiers or timelines); (2) redefine the study’s scope to address a specific, unanswered question (e.g., biomarkers predicting response); and (3) substantially expand the cohort or include a control arm. However, given the current state, these revisions seem unlikely to salvage its contribution.

References

  1. Haque, E., Muhsen, I. N., Esmail, A., Umoru, G., Mylavarapu, C., Ajewole, V. B., & Abdelrahim, M. (2022). Case report: Efficacy and safety of regorafenib plus fluorouracil combination therapy in the treatment of refractory metastatic colorectal cancer. Frontiers in Oncology, 12, 992455. https://doi.org/10.3389/fonc.2022.992455
  2. Lin, C. Y., Lin, T. H., Chen, C. C., Chen, M. C., & Chen, C. P. (2018). Combination chemotherapy with regorafenib in metastatic colorectal cancer treatment: A single center, retrospective study. PLoS One, 13(1), e0190497. https://doi.org/10.1371/journal.pone.0190497
  3. Marks, E. I., Tan, C., Zhang, J., Zhou, L., Yang, Z., Scicchitano, A., & El-Deiry, W. S. (2015). Regorafenib with a fluoropyrimidine for metastatic colorectal cancer after progression on multiple 5-FU-containing combination therapies and regorafenib monotherapy. Cancer Biology & Therapy, 16(12), 1710-1719. https://doi.org/10.1080/15384047.2015.1113355